# Ocean acidification decreases grazing pressure but alters morphological structure in a dominant coastal seaweed

**Alexandra Kinnby**[ORCID]**\*, Joel C. B. White, Gunilla B. Toth, Henrik Pavia**

Tjärnö Marine Laboratory, Department of Marine Sciences, University of Gothenburg, Strömstad, Sweden

\* alexandra.kinnby@marine.gu.se

## Abstract

Ocean acidification driven by anthropogenic climate change is causing a global decrease in pH, which is projected to be 0.4 units lower in coastal shallow waters by the year 2100. Previous studies have shown that seaweeds grown under such conditions may alter their growth and photosynthetic capacity. It is not clear how such alterations might impact interactions between seaweed and herbivores, *e.g.* through changes in feeding rates, nutritional value, or defense levels. Changes in seaweeds are particularly important for coastal food webs, as they are key primary producers and often habitat-forming species. We cultured the habitat-forming brown seaweed *Fucus vesiculosus* for 30 days in projected future $pCO_2$ (1100 µatm) with genetically identical controls in ambient $pCO_2$ (400 µatm). Thereafter the macroalgae were exposed to grazing by *Littorina littorea*, acclimated to the relevant $pCO_2$-treatment. We found increased growth (measured as surface area increase), decreased tissue strength in a tensile strength test, and decreased chemical defense (phlorotannins) levels in seaweeds exposed to high $pCO_2$-levels. The herbivores exposed to elevated $pCO_2$-levels showed improved condition index, decreased consumption, but no significant change in feeding preference. Fucoid seaweeds such as *F. vesiculosus* play important ecological roles in coastal habitats and are often foundation species, with a key role for ecosystem structure and function. The change in surface area and associated decrease in breaking force, as demonstrated by our results, indicate that *F. vesiculosus* grown under elevated levels of $pCO_2$ may acquire an altered morphology and reduced tissue strength. This, together with increased wave energy in coastal ecosystems due to climate change, could have detrimental effects by reducing both habitat and food availability for herbivores.

**Data Availability Statement:** All relevant data are within the manuscript and its Supporting Information files.

## Introduction

Ocean acidification (OA) is the decrease in pH caused by the absorption of atmospheric $CO_2$ into the surface of the oceans [1]. The majority of dissolved $CO_2$ concentrates above the thermocline, generating an estimated drop in pH to 7.7 [2] or 0.4 units [1,3,4] by year 2100 in open ocean surface waters and the entire water column in the shallow coastal waters [5]. Thus, coastal ecosystems and the organisms that live there are expected to be among the most

**Funding:** This work was funded by the Swedish Research Council VR and Formas through a Linnaeus grant to the Centre for Marine Evolutionary Biology (CeMEB; http://cemeb. science.gu.se 217-2008-1719 awarded to Henrik Pavia, and by Rådman och Fru Ernst Collianders stiftelse för välgörande ändamål awarded to Alexandra Kinnby. The funders had no role in study design, data collection and analysis, decision to publish, or preparation of the manuscript.

**Competing interests:** The authors have declared that no competing interests exist.

impacted by OA. Seaweeds are key habitat-forming primary producers that support high bio-diversity in coastal areas [6] and therefore their responses to OA may have impacts throughout the ecosystem. Seaweeds primarily use $CO_2$, and most species also use $HCO_3^-$, for carbon fixation and growth, and may therefore benefit from the increase in available carbon caused by OA [7]. A growing number of studies have, however, shown that OA can have positive, neutral, or negative direct effects on basic performance traits such as growth and photosynthesis of seaweeds [*e.g.* 7,8], and these effects may differ between life stages of a species, as well as between closely related species [9–11].

Aside from effects on basic performance traits, OA can also impact both primary and secondary metabolism in seaweeds, sometimes resulting in higher carbon to nitrogen (C:N) and carbon to phosphorous (C:P) ratios [but see 8,12–14], which indicate a change in the nutritional content of the seaweed tissue. Increase in carbon availability for seaweeds generally results in decreased protein content [*e.g.* 15–19], and either increased [17,18] or decreased [20] levels of fatty acids. Furthermore, the content of secondary metabolites, such as the grazing deterrent dimethylsulfoniopropionate (DMSP) in green seaweeds, has been shown to increase in response to elevated $pCO_2$ levels [15]. In brown seaweeds phlorotannins (polyphenolic compounds) are ubiquitous metabolites that can occur in high concentrations, especially in fucoid species (*Fucales*). Phlorotannins have multiple functions *e.g.* as defense against UV-radiation and defense compounds against gastropod grazing [21,22]. To our knowledge, only two studies have investigated the effects of OA on phlorotannin production in brown seaweeds, with mixed results [9,11]. Olischläger *et al.* [9] found no effect on phlorotannin production in the kelp *Laminaria hyperborea* when grown under 700 μatm, while Swanson & Fox [11] found increased phlorotannin production in *Saccharina latissima* but not *Nereocystis leutkeana* when exposed to 3000 μatm $pCO_2$.

The nutritional and defensive characteristics of seaweeds are critical traits in ecological interactions since they affect the growth and fecundity of herbivores [23,24]. Therefore, apart from direct effects on the physiology and biochemical content, OA may also have indirect effects on macroalgae through interactions with grazers. A decrease in the nutritional value and increase in deterrent defense metabolites under OA may lower the palatability of macroalgae to grazers [*e.g.* 16,25,26]. This may, however, also lead to an increase in the *per capita* grazing pressure through compensatory grazing if less nutritious food is available [*e.g.* 16]. Grazing may also be altered by direct effects of OA on the herbivore, *e.g.* through changes in respiration or behavior [*e.g.* 27,28]. Bibby *et al.* [27] showed that the snail *Littorina littorea* had noticeable reductions in both metabolic rate and induced defense (shell formation), which increased the avoidance behavior of the snails and could in turn affect their interactions with other species. Additionally, Young *et al.* [28] found that the grazing rate of a snail (*Lacuna vincta*) decreased when it was exposed to elevated $pCO_2$, regardless of the effects of $pCO_2$ on the seaweeds (*Ulva* spp.) that the snail was grazing on.

In temperate coastal ecosystems, fucoids are dominant habitat-forming seaweeds that provide shelter, habitat, and food for other organisms [29]. The presence of fucoids is associated with a local increase in species abundance and diversity [30], but there is no consensus how OA will affect the adult stage of associated species [but see *e.g.* 31 for effects on early life-stages]. Since many fucoids have an active uptake of bicarbonate [32], which is abundant in seawater (up to 91% [7]), it has been suggested that they should not increase growth in response to increased $pCO_2$ since they may not be carbon limited [33]. We are only aware of two studies that investigate potential indirect effects of OA on fucoids through changes in interactions with herbivores [34,35]. One of these studies found that the herbivore *Littorina obtusata* consumed more of *Ascophyllum nodosum* under OA conditions, albeit this difference was not statistically different [35]. In contrast, the other study showed no effect of decreased

pH on the interaction between herbivores and *F. vesiculosus* [34]. This relative lack of literature is surprising, considering the abundance of fucoids. Given the high phlorotannin content found in many fucoids, the effect of OA on phlorotannin production may also alter the interaction between seaweed and herbivore, but this has, to our knowledge, not yet been investigated.

The overall aim of the present study was to examine potential direct effects of OA on the fucoid *F. vesiculosus*, as well as indirect effects on the seaweed through changes in its interactions with the gastropod grazer *L. littorea*, both common species along coasts in the North Atlantic. We conducted manipulative experiments to determine how the growth rate, photosynthesis, carbon and nitrogen content, as well as chemical defense (phlorotannin content), and breaking strength of *F. vesiculosus* will be affected by increasing $pCO_2$ levels in the future. Furthermore, we also tested the effect of elevated $pCO_2$ on consumption, feeding preference, and condition index of *L. littorea*.

## Materials and methods

### Experimental design

Sixty individuals of *F. vesiculosus* were collected from the west coast of Sweden, in July 2018 and kept at Tjärnö Marine Laboratory (TML, 58˚52'36.4"N 11˚6'42.84"E) under ambient conditions (Table 1) for 7 days to acclimatize. Due to the small tidal range in the area, *F. vesiculosus* in western Sweden can be submerged for long time periods depending on prevailing weather conditions (personal observations, A. Kinnby), hence the algae were kept under water throughout the experiment. The experiment was performed in a greenhouse with natural lighting (natural light cycle 18:6 h, L:D). After the acclimation period, each seaweed was split into one experimental thallus and one control thallus, placed in separate 1L aquaria (a total of 120 aquaria, *i.e.* n = 60) with constant seawater flow from header tanks (4 per treatment, n = 15). Control thalli were maintained at ambient $pCO_2$ (400 μatm) while experimental thalli were exposed to gradually increasing (~100 μatm/day) $CO_2$ until a $pCO_2$ of 1100 μatm was reached 7 days later (corresponding to the projected value at the end of this century [36]). 360 individuals of similar sized *L. littorea* were also collected and exposed to the same conditions as the seaweed, *i.e.* 180 snails were exposed to ambient water and 180 snails were exposed to treatment water in separate tanks from the seaweed thalli (these snails were used in a grazing experiment described below). The header tanks were aerated with either ambient atmospheric air ($pCO_2$ of 400 ppm) or $CO_2$-enriched air controlled by solenoid valves and pH-computers (Aqua Medic) to provide a final $pCO_2$ of 1100 μatm. The $pCO_2$ was monitored daily with LI-850 $CO_2/H_2O$ Gas Analyzer (Li-COR). The $CO_2$ analyzer was calibrated with custom mixed gas, 970 ppm (Linde Gas AB, Sweden). Filtered seawater (5 μm) flow was constant at 0.3 L/min in each aquarium throughout the experiment. Salinity, temperature, $pCO_2$, and $pH_{NBS}$ were measured in the 1L aquaria. pH was recorded using HANNA instruments pH electrode HALO probe (HI-1102) calibrated with NBS pH 4.01, 7.01, and 10.01 standards (HANNA

**Table 1. Seawater chemistry of experimental treatments; partial pressure of $CO_2$ ($pCO_2$), $pH_{NBS}$, salinity, and temperature were measured twice a week.**

|  | pCO₂ (μatm) | pH_NBS | pH_T | A_T (μmolkg⁻¹) | Salinity (PSU) | Temperature (˚C) |
|---|---|---|---|---|---|---|
| **Ambient** | 400 ± 47 | 8.04 ± 0.03 | 8.05 | 2258 | 32 ± 0.8 | 15 ± 1 |
| **Treatment** | 1100 ± 61 | 7.64 ± 0.04 | 7.66 | 2258 | 32 ± 0.8 | 15 ± 1 |

Total alkalinity was estimated from salinity using long-term salinity:alkalinity relationship data for this location (r = 0.94) and $pH_T$ was calculated from the temperature, salinity, $pCO_2$, and total alkalinity using CO2calc. Data are averages (SD), n = 8.

instruments) before each measurement. Total alkalinity was estimated from salinity using long-term salinity:alkalinity relationship data for this location (r = 0.94; data obtained from SMHI https://www.smhi.se/data/oceanografi/datavardskap-oceanografi-och-marinbiologi/sharkweb) [37] and $pH_T$ was calculated from the temperature, salinity, $pCO_2$, and total alkalinity using CO2calc [38; Table 1].

All seaweed thalli were weighed fresh (n = 60) and photographed (for area (n = 60) measurements using Image J [39]) at the beginning and end of the 30-day experiment. At the end of the experiment the efficiency of photosystems II (Fv/Fm and P- index, n = 60 for both measurements) were measured in the new tissue formed during the experiment. The tissue was dark-adapted for 10 minutes, the fiber optics were held at a fixed 10 mm distance from the algae, and measurements were taken with a PAM (pulse amplitude-modulated fluorometer; Walz, Effeltrich, Germany). Breaking strength was measured by securing the seaweed to a dynamometer (Lutron FG-5020; Taiwan) such that only one apical tip was being strained, and increasing the strain until the thallus broke (n = 10). Thus, measuring breaking strength on tissue that was formed during the experiment. Following this, apical tissue samples were frozen (-60˚C) for further elemental and phlorotannin analysis, the remaining tips were used in the consumption and preference experiment with *L. littorea* (see below).

## Phlorotannin analysis

For phlorotannin analysis, the frozen samples (n = 60) were freeze-dried, homogenized to a fine powder, and 10 mg of each sample was extracted in 60% acetone. Total phlorotannin content was quantified colorimetrically using the Folin-Ciocalteu method [40], with phloroglucinol (1,3,5-trihydroxybenzene, art. 7069; Merck, Darmstadt, Germany) as a standard. Results are presented as % dw (dry weight).

## Elemental analysis

For the determination of carbon (C) and nitrogen (N) content the frozen seaweed tissue was freeze-dried and homogenized to a fine powder and weighed to the nearest 0.01 mg. The total tissue C and N content, as well as δ13C and δ15N of the samples (n = 60) were analyzed with an elemental analyzer (ANCA-GSL, Sercon Ltd., Crewe, UK) coupled to an isotope ratio mass spectrometer (20–22, Sercon Ltd., Crewe, UK).

## Consumption and feeding preference of *Littorina littorea*

The palatability of the *F. vesiculosus* thalli grown in ambient and elevated $pCO_2$ during 30 days was measured in two-choice feeding trials using starved *L. littorea* as the grazer. The feeding experiment was performed using a total of 120 containers (200 mL) with constant seawater flow of ambient $pCO_2$ (400 μatm). In each container two similarly sized apical pieces of *F. vesiculosus* were placed (0.50 ± 0.019 g mean ± SD), one piece from the ambient $pCO_2$ treatment and one from the elevated $pCO_2$ treatment; as both pieces came from the same thallus they were genetically identical. Six individuals of *L. littorea*, exposed to either ambient or elevated $pCO_2$ were placed in half of the containers (n = 30, *i.e.* a total of 60 containers with herbivores). To control for autogenic changes in mass (*i.e.* growth) during the experiment that was not caused by the grazing of the snails, each container with seaweed pieces and herbivores were paired with an identical control container without herbivores containing similarly sized apical pieces from the same genetic individual of seaweed. The wet weight of all seaweed pieces was determined at the start and at the end of the 24-hour experiment by using a standard blotting procedure, and the wet-weight change of each seaweed piece was calculated by subtracting the weight at the end of the experiment from the starting weight. The consumption of the snails

exposed to different $pCO_2$-levels was determined by calculating the total change in weight between initial and post-grazing weights for each container and subtracting the weight change in the autogenic control containers. To study feeding preference of herbivores exposed to different $pCO_2$-levels, the difference between weight changes of the two seaweed pieces in each container was calculated by subtracting the wet weight change of the seaweed exposed to elevated $pCO_2$ from the weight change of the seaweed piece exposed to ambient $pCO_2$ [41].

## Condition index for *Littorina littorea*

Following the feeding preference experiment all snails were euthanized by freezing at -20˚C. To assess whether the elevated $pCO_2$ treatment had affected the physiological status of the snails a condition index was calculated, where a higher condition index is a sign of a healthier individual [35,42]. Snails were thawed and weighed whole; following this the shell was weighed alone. The dry weight of the soft tissue was obtained by weighing the body after drying for 48 hours at 50˚C. The condition index was derived from the weights according to the following formula:

$$= (100 * dry\ tissue\ weight)/(whole\ weight - shell\ weight)$$

## Data analysis

The seaweed response variables, *i.e.* growth (% increase in area and weight), breaking strength, efficiency of photosystem II (Fv/Fm and P-index), as well as phlorotannin, and nutritional content were all statistically analyzed with mixed model ANOVAs with $pCO_2$ treatment as a fixed factor and header tank as a random factor nested within $pCO_2$ treatment. However, since header tank was non-significant ($p > 0.40$ for all variables, the mean square for this factor was pooled with the residual mean square and paired t-tests were used to determine if there was a significant difference between the seaweed in ambient and elevated levels of $pCO_2$. Paired t-tests were used because every treatment thallus was paired with a genetically identical control thallus. Before analysis the data for each response variable was checked and found to meet the assumptions of normality. To investigate if there was a difference in condition index between the snails exposed to ambient and treatment water a t-test was performed. The condition index data was not normally distributed, hence a Mann Whitney U-test was run. The consumption of herbivores exposed to ambient and elevated $pCO_2$ was analyzed with a t-test. Preference for seaweed grown under the different $pCO_2$ conditions was evaluated by comparing the difference in weight change between the seaweed pieces kept with the herbivores and their respective autogenic controls with two separate paired t-tests; one each for herbivores exposed to ambient and elevated $pCO_2$. A significantly lower difference in wet weight change for seaweed pieces kept with herbivores compared to autogenic controls will indicate a preference for feeding on the control seaweed [41]. All analyses were performed in RStudio (version 1.0.136).

## Results

The seaweed thalli exposed to elevated levels of $pCO_2$ grew significantly more than the thalli exposed to ambient $pCO_2$ when growth was measured as increase in surface area of the seaweed (Fig 1A; Table 2). On average, growth rates under elevated $pCO_2$ were 34% higher than growth under ambient conditions. However, thallus weight did not differ significantly between the two treatments (Fig 1B; Table 2). We found a significant difference in the force needed to break the seaweed tissue in the control group compared to the seaweeds exposed to elevated $pCO_2$; thalli from the treatment group were 57% weaker than those in the control group

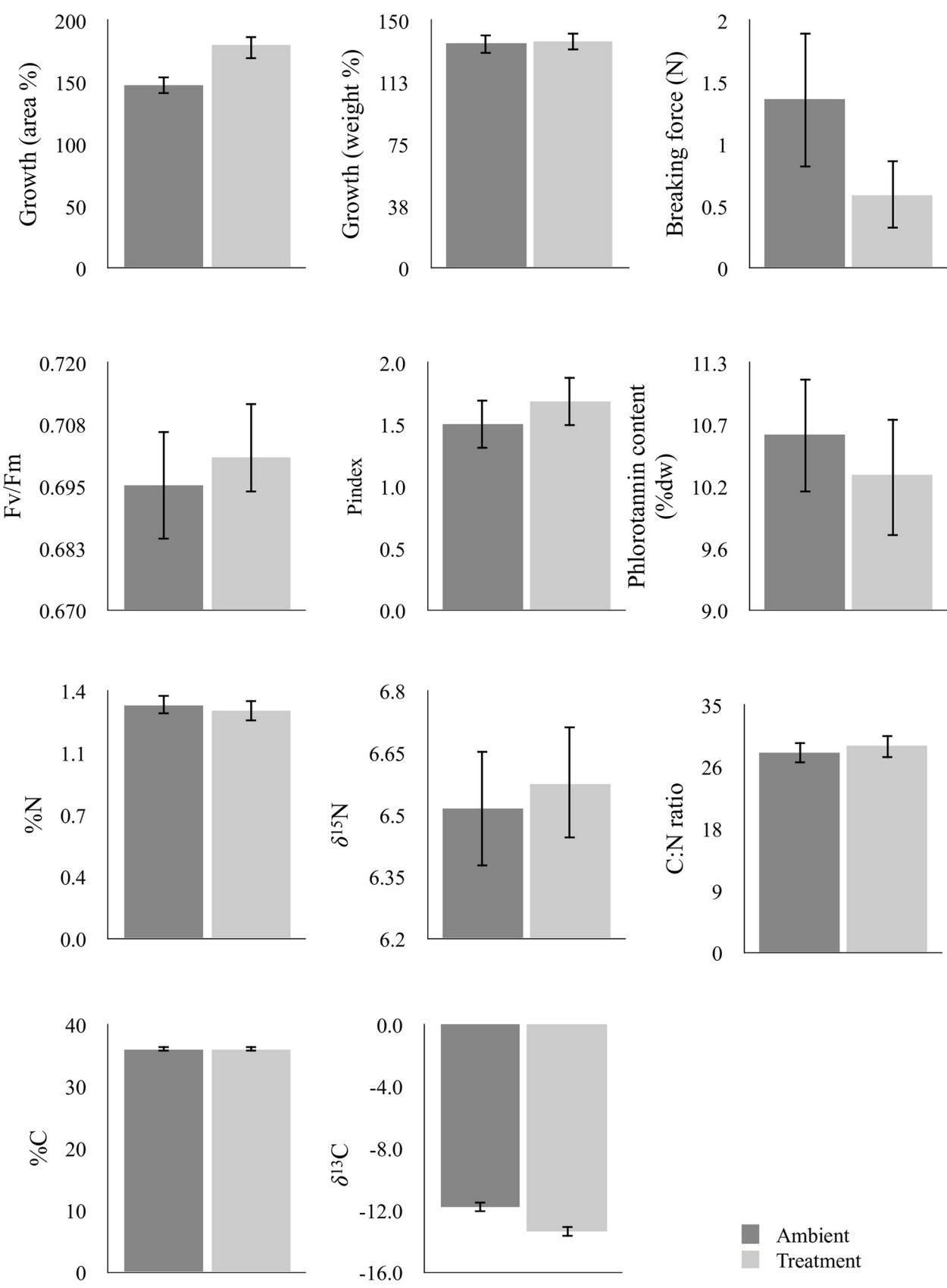

**Fig 1. Effects on response variables of *Fucus vesiculosus* grown under ambient (400 µatm) and elevated (1100 µatm) $pCO_2$ for 30 days.** Values are means ± 95% CI, n = 60 for a-j and n = 10 for k. Response variables measured as a) Growth measured % increase in area (n = 60), b) growth measured as % increase in weight (n = 60), c) breaking force (N) (n = 10), d) efficiency of photosystem II (Fv/Fm) (n = 60), e) efficiency of photosystem II (P index) (n = 60), f) Phlorotannin content (%dw) (n = 60), g) %Nitrogen (n = 60), h) $\delta^{15}N$ (n = 60), i) C:N ratio (n = 60), j) % Carbon (n = 60), and k) $\delta13C$ (n = 60).

(Fig 1C; Table 2). We found no statistically significant differences in the efficiency of photosystem II measured as Fv/Fm and P-index (Fig 1D and 1E; Table 2).

The results of chemical analyses showed that there was a statistically significant decrease (3%) in phlorotannin content between the tips of thalli exposed to elevated levels of $pCO_2$ and those being exposed to ambient levels (Fig 1F; Table 2). There were no statistically significant differences in C or N tissue content, nor in the C:N ratio. However, the stable carbon isotope ($\delta^{13}C$) content of *F. vesiculosus* was significantly reduced when exposed to elevated $pCO_2$; the $\delta^{13}C$ values decreased to -13% under elevated $pCO_2$ levels. Tissue $\delta^{15}N$ was not significantly changed when exposed to elevated levels of $pCO_2$ (Fig 1G–1K; Table 2).

The mean condition index of *L. littorea* was 28.9% higher in snails that had been exposed to high $pCO_2$-levels than those exposed to ambient $pCO_2$-levels (U = 13155, z-score = -2.60, p = 0.0093; Fig 2). Despite having a higher condition index, the herbivores exposed to high $pCO_2$ levels consumed 37.5% less than the snails exposed to ambient $pCO_2$ (t-test, $t_{58} = 2.67$, p = 0.0098, Fig 2). However, the herbivores did not show a statistically significant difference in preference based on the experimental treatment of the seaweed, regardless if the herbivores had been exposed to ambient $pCO_2$ (paired t-test, $t_{29} = -0.517$, p = 0.609) or elevated $pCO_2$ (paired t-test, $t_{29} = -0.584$, p = 0.564).

## Discussion

Seaweeds play important ecological roles in coastal habitats and are often foundation species, with a key role for ecosystem structure and function. Hence, it is important to understand how seaweeds will be directly affected by changes in their environment, and also if these changes will alter seaweed interactions with other species. Here, we show that elevated $pCO_2$-levels increased the thallus area, decreased the phlorotannin content, and reduced the breaking strength of *F. vesiculosus*. This may result in that the seaweeds become less robust in field

**Table 2. Summary of effects of ambient (400 µatm) and elevated (1100 µatm) levels of $pCO_2$ on *Fucus vesiculosus* measured as ten responses.**

| Response variable | p-value | t-value | Df | Mean (400ppm) | 95%CI (400ppm) | Mean (1100ppm) | 95%CI (1100ppm) |
|---|---|---|---|---|---|---|---|
| Growth: area (%) | **9.2e-07** | -5.48 | 59 | 147.5 | 6.54 | 180.1 | 11.47 |
| Growth: weight (%) | 0.767 | -0.30 | 59 | 136.0 | 5.91 | 136.7 | 4.68 |
| Breaking force (N) | **0.032** | 2.33 | 18 | 1.36 | 0.54 | 0.59 | 0.27 |
| Photosystem II (Fv/Fm) | 0.407 | -0.83 | 59 | 0.695 | 0.011 | 0.701 | 0.007 |
| Photosystem II (P index) | 0.147 | -1.47 | 59 | 1.5 | 0.20 | 1.7 | 0.19 |
| Phlorotannin content (% dw) | **0.030** | 2.22 | 59 | 10.6 | 0.53 | 10.3 | 0.57 |
| % Nitrogen | 0.379 | 0.89 | 59 | 1.32 | 0.055 | 1.29 | 0.066 |
| % Carbon | 0.865 | 0.17 | 59 | 36.04 | 0.43 | 36.00 | 0.37 |
| C:N | 0.236 | -1.20 | 59 | 28.2 | 1.44 | 29.3 | 1.80 |
| $\delta^{13}C$ | **4.762e-09** | 6.85 | 59 | -11.77 | 0.32 | -13.33 | 0.41 |
| $\delta^{15}N$ | 0.315 | -1.01 | 59 | 6.51 | 0.14 | 6.57 | 0.13 |

P-values and corresponding t-values and degrees of freedom of paired t-tests are reported for the analyses of all response variables as well as means and 95% confidence intervals. Values in bold denote statistically significant values.

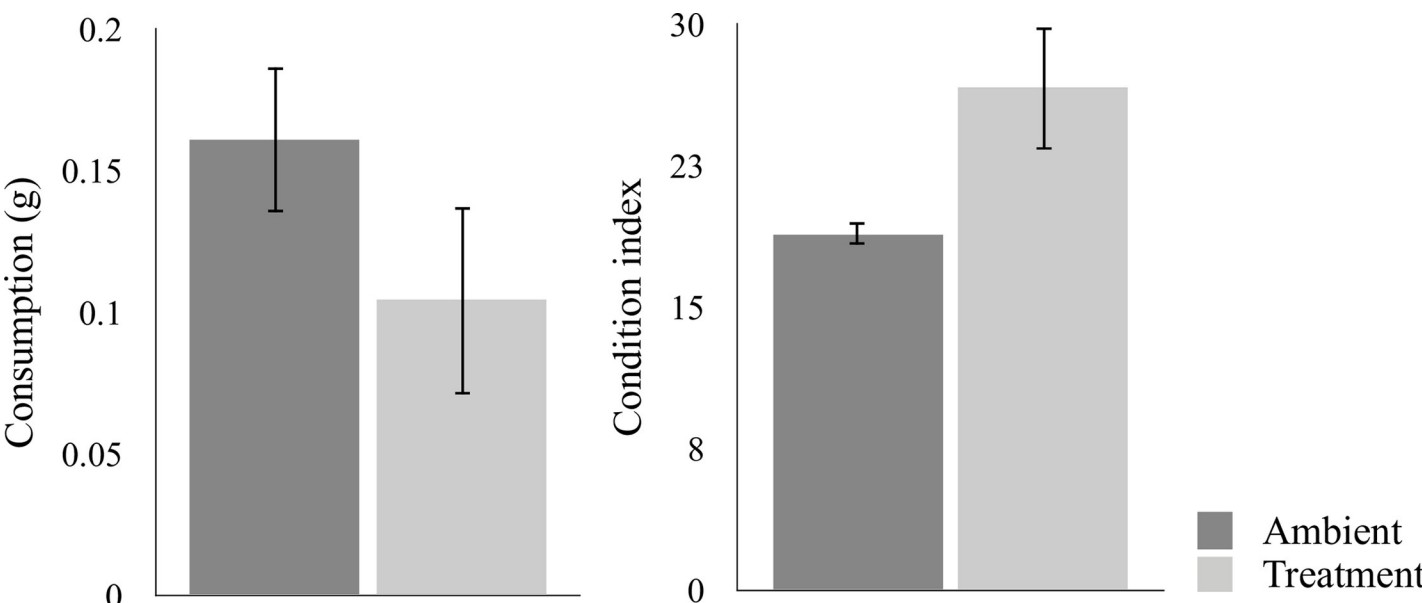

**Fig 2.** a) Consumption of the alga *Fucus vesiculosus* by the gastropod *Littorina littorea*, exposed to ambient (400 μatm) and elevated (1100 μatm) $pCO_2$ for 30 days. b) Condition index for individuals of *L. littorea* following the grazing experiment. Values are means ± 95% CI.

conditions. This could lead to an overall loss of seaweed coverage which in turn is likely to affect all the organisms that live in, or consume, this seaweed. The condition index of the snails increased under exposure to elevated levels of $pCO_2$, but the consumption decreased and we saw no significant effect of treatment on the palatability of *F. vesiculosus* thalli.

## Effects on growth

Increased $pCO_2$ had no significant effect on the weight change of *F. vesiculosus*, but significantly increased the thallus surface area. Previous studies with *F. vesiculosus* have reported unaltered [43,44] or reduced [45] growth, measured as wet weight, under elevated $pCO_2$-levels. Graiff *et al*. [46], however, reported a tendency (not statistically significant) of higher growth of *F. vesiculosus*, measured both as wet weight and length in apical tips, at elevated $pCO_2$ levels. These different experimental results suggest that other factors interact with $pCO_2$ to determine growth, *e.g.* seasonality or the life-cycle stage (age) of the seaweed, or genetic differences due to local adaptation among the populations used in the different studies. Such genetic differences in phlorotannin production and growth were recently demonstrated among *F. vesiculosus* populations at distances less than 100 km [47].

## Effects on breaking strength

The combination of a significantly larger thallus with no effect on the weight of *F. vesiculosus* under enhanced $CO_2$ conditions found in the present study strongly indicates a decrease in tissue density, which is corroborated by the drastic (57%) decrease in breaking strength of the thallus. To our knowledge, this is the first time such an effect of increased $pCO_2$ levels is reported for seaweeds, and it parallels findings in developing seaweed spores and terrestrial plants. For example, Guenther *et al*. [48] found that reduced pH delayed spore attachment in two different red algae, while Pretzsch *et al*. [49] documented an increase in growth rate, attributed to increasing $CO_2$ levels, among tree species in central Europe between 1960 and 2014. They also showed that this increase in growth was coupled with a decrease in tissue

density and/or strength. This leaves forests more vulnerable to the increased weather variability that is also associated with climate change [49,50]. Our results suggest that similar effects may also be present in coastal marine systems. The reduced breaking strength could make seaweeds more vulnerable to storms and wave action, which are projected to become more frequent as the climate changes [51,52]. Increased vulnerability will likely reduce the role *F. vesiculosus* plays in the nearshore ecosystem, with negative impacts on species that rely on this seaweed for food or habitat.

## Effects on photosynthesis

We observed no significant differences in the Fv/Fm ratio or P-index in our experiment, suggesting that *F. vesiculosus* does not increase the maximum quantum efficiency of photosystem II or sample vitality in response to elevated $pCO_2$. This follows the findings of Fernández and colleagues [53], who demonstrated that the increased carbon availability from increased $pCO_2$-levels had no effect on photosynthesis. Seaweeds in general acquire carbon by passive diffusion of $CO_2$ and active transportation of bicarbonate; as the concentration of $CO_2$ rises the amount of passively diffusing $CO_2$ potentially also rises, reducing the seaweed's reliance on active transport proteins, and potentially allowing more energy to be allocated for growth [54]. Increased uptake of $CO_2$ coincides with a decrease in tissue $\delta^{13}C$ [8]. In this study we found that the $\delta^{13}C$ decreased from -11.77% to -13.33% when seaweeds were exposed to increased levels of $CO_2$, which indicates a transition away from active intake of bicarbonate towards passive uptake of $CO_2$. A similar change was previously documented in both *Gracillaria* sp. and *Ulva* sp.,[54], as well as in *Lomentaria australis* where an increase in growth and decrease in $\delta^{13}C$ were hypothesized to indicate a transition away from a more costly CCM (carbon dioxide-concentrating mechanism) [8]. In our study, however, this did not translate to an increase in biomass as we did not find any significant differences in the weight gain of the seaweeds exposed to different levels of $CO_2$. *Fucus vesiculosus* can use two parallel $CO_2$ pathways for photosynthesis, both directly taking up carbon from their environment and also storing it as an organic intermediate for use when other carbon in less available [55], suggesting that under normal circumstances *F. vesiculosus* plants are most likely not carbon-limited. This, together with the fact that *F. vesiculosus* used in our experiment were constantly submerged (which is common due to the low tidal range along the Swedish west coast) may explain the lack of effect from increased $pCO_2$ on growth measured as weight gain.

## Effects on elemental and phlorotannin content

In terrestrial plants, increased atmospheric $CO_2$ has been shown to increase C:N ratios as well as lead to an accumulation of phenolic compounds, such as tannins, affecting the consumption and growth rates of grazers [56,57]. However, we found no significant differences in C or N tissue content, nor in the C:N ratio. In contrast, Gutow *et al.*, [45] showed that elevated levels of $pCO_2$ (700 µatm) decreased the C:N ratio of *F. vesiculosus*. Studies on the effects of increased $CO_2$ on phenolic compounds in marine macrophytes are few and only one previous study on kelp species found that elevated $CO_2$ leads to increased levels of phlorotannins [11]. By contrast, marine vascular plants have been shown to reduce phenolic acid production under increased $CO_2$ conditions [58], which aligns with our results on *F. vesiculosus* showing slightly lower phlorotannin content in apical tips exposed to elevated $pCO_2$-levels.

## Effects on interactions with a grazer

Despite finding a somewhat lower phlorotannin content in seaweeds exposed to elevated $pCO_2$, we did not find a difference in grazing preference of the gastropod *L. littorea* between

seaweeds from the different treatments. However, snails exposed to elevated $pCO_2$ generally consumed less than those exposed to ambient conditions, regardless of which food type they were offered. Reduced consumption by snails at increased $pCO_2$ levels could indicate easier ingestion and digestion of food, or decreased activity of the grazer (and therefore decreased caloric requirements), in line with previous research on *L. littorea* [27] and other marine invertebrates [59] showing reduced metabolic rates at increased $pCO_2$ levels. The snails exposed to elevated levels of $pCO_2$ had a higher condition index than snails exposed to ambient conditions. This combination of results is surprising, as a decreased consumption would be expected to result in a drop in condition index. Increased condition index, *i.e.* a higher dry to wet weight ratio of the soft tissue, could indicate more accumulation of tissue, *i.e.* increased growth, but also possibly failure to osmoregulate or other associated physiological problems. In summary, the results from the feeding experiment in the present study suggest that there are direct effects of increased $pCO_2$ on herbivores and their consumption of seaweeds, but any indirect effects mediated by changes the palatability of the seaweeds are harder to discern.

## Conclusion

In conclusion, our study shows that under OA conditions the habitat forming seaweed *F. vesiculosus* increases growth by thallus area, reduces reliance on active carbon uptake, shows a slight decrease in phlorotannin content and a drastic reduction in breaking strength. At the same time the herbivore *L. littorea* seems to tolerate increased $pCO_2$ with an increased condition index even as they reduce their consumption of seaweeds. Reduced consumption for the herbivore suggests that the seaweed could gain some ecological benefits under OA. However, our most unanticipated finding–that the seaweed could become more vulnerable to physical forces under OA because of a significantly reduced breaking strength–could result in loss of seaweed biomass due to increased storm events that are associated with climate change. This might in turn have implications for the future community structure of shallow coastal areas under OA.

## Supporting information

**S1 File.**
(XLSX)

## Acknowledgments

We are grateful to Gunnar Cervin (University of Gothenburg) for help with the experiments and Kerstin Johannesson (University of Gothenburg) for valuable comments on the manuscript.

## Author Contributions

**Conceptualization:** Alexandra Kinnby, Henrik Pavia.

**Data curation:** Alexandra Kinnby.

**Formal analysis:** Alexandra Kinnby.

**Funding acquisition:** Alexandra Kinnby, Henrik Pavia.

**Investigation:** Alexandra Kinnby, Joel C. B. White.

**Methodology:** Alexandra Kinnby, Joel C. B. White.

**Project administration:** Alexandra Kinnby.

**Supervision:** Gunilla B. Toth, Henrik Pavia.

**Visualization:** Alexandra Kinnby.

**Writing – original draft:** Alexandra Kinnby.

**Writing – review & editing:** Alexandra Kinnby, Joel C. B. White, Gunilla B. Toth, Henrik Pavia.

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
