## [Decision Letter · Decision Letter 0]

23 Oct 2020

PONE-D-20-25580

Increased thallus area and decreased breaking strength in a habitat forming seaweed under ocean acidification

PLOS ONE

Dear Alexandra Kinnby, 

Thank you for submitting your manuscript to PLOS ONE. After careful consideration, we feel that it has merit but does not fully meet PLOS ONE’s publication criteria as it currently stands. Therefore, we invite you to submit a revised version of the manuscript that addresses the points raised during the review process.

We look forward to receiving your revised manuscript.

Kind regards,

Christopher Edward Cornwall, Ph.D.

Academic Editor

PLOS ONE

Journal Requirements:

2. In your Methods section, please provide additional location information of the collection sites, including geographic coordinates for the data set if available.

3. In your Methods section, please provide additional information regarding the permits you obtained for the work. Please ensure you have included the full name of the authority that approved the collection sites access and, if no permits were required, a brief statement explaining why.

Additional Editor Comments (if provided):

I have received the reports of two different reviewers back. They both support the work and offer minor comments that should be addressed in the revisions. I have also read the manuscript, and I consider that the authors should pay particular attention to the comments from reviewer 2 regarding the numbers of tanks etc. This is extremely important in determining the validity of the replication here. Please indicate how many header tanks, experimental tanks etc. per treatment, and how any interdependence was dealt with in the model. I cannot currently see this clearly described.

Reviewers' comments:

Reviewer's Responses to Questions

**Comments to the Author**

1. Is the manuscript technically sound, and do the data support the conclusions?

Reviewer #1: Yes

Reviewer #2: Yes

2. Has the statistical analysis been performed appropriately and rigorously? 

Reviewer #1: Yes

Reviewer #2: Yes

3. Have the authors made all data underlying the findings in their manuscript fully available?

Reviewer #1: Yes

Reviewer #2: Yes

4. Is the manuscript presented in an intelligible fashion and written in standard English?

Reviewer #1: Yes

Reviewer #2: Yes

5. Review Comments to the Author

Reviewer #1: Kinnby et al exposed Fucus vesiculosus and Littorina littorea to ocean acidification conditions predicted for the year 2100 to look at the direct and indirect effects on organismal response and grazer interaction, respectively. Kelp growth, breaking strength, phlorotannin production were affected by OA, while grazer preference was unaffected but grazers decreased consumption and improved condition index. The manuscript provides new information on ecologically important species performance and interaction in future oceans. Overall, the manuscript is presented and written in a straightforward manner. However, more details are needed in the methods section. As the authors point out, there are discrepancies in the OA literature, much of which likely comes from different experimental setups and techniques, so it is imperative to provide as much information as possible. Provided the authors amend the methods section, I think this manuscript would be a good addition for Plos One and add to our understanding of OA effects on different species. I have performed the review with track changes in Word, pointing out missing details that should be added, also editing some text to help clarity or flow, which the authors can use at their discretion.

Reviewer #2: The manuscript provides interesting and useful data on the response of habitat forming seaweeds to ocean acidification. With some minor improvements i consider the manuscript warrants publication. The main issue i have is the discussion in which i think the interesting results could be better placed in the context of the broader literature and structured to better highlight the impacts of the key findings.

Specific comments:

Abstract

Line 34

change “available” to “availability”

Introduction

Line 41

in surface and coastal waters? A little confusing with the wording, I think you mean to suggest that its surface of the open ocean but the coastal waters will be all the water column due to it being shallow. Might help to have a reword for clarity.

Line 42-43

The sentence that says coastal organisms the most impacted. I suggest saying one of the most impacted.

Line 55-56

See also van der Loos et al 2019 for no change.

Line 83

Change “Temporal” to “Temperate”

Line 90-94

This sentence is a little confusing – suggest a slight reword.

Line 94-95

I have noticed a little bit of repeatability about the importance of fucoids/seaweeds in general in coastal systems – e.g. Lines 83-84 and 43-44. I suggest going through the introduction and reducing repetition of this point wherever possible.

Line 96-99

Suggest framing this point as important rather than interesting as it is both interesting and important!

Line 101-107

Are both these species common globally or just in the region you work? As written, it suggests globally.

Methods

Line 128-129

Could you provide frequency and total number of measurements? Also what equipment was used to measure these parameters? This info needs to be included.

Table 1

Please provide standard error in the table for the measured parameters. Was AT estimated from only pCO2 and salinity or also pH and temp? Needs to be clarified in the caption/text. Was pH measured on the NBS scale only? Ideally seawater pH should be measured on the Total scale. For comparison to other studies could the pH also be reported on the Total scale (if not measured) using estimates from CO2Calc and this stated?

Line 135

All seaweed thalli? Same question for phlorotannin and elemental analysis and breaking strength. Please clarify and include n per treatment for each response variable.

Line 140

Could you include make and model of the equipment.

Line 148

Change to “as a standard”.

Line 157-177

I found it a little hard to follow whether the snails exposed to different pCO2 were placed in separate containers and then also if not how this was accounted for in the data analysis? Some more clarity would be useful.

Line 179

For readers not familiar with condition index could you provide a brief description of whether higher or lower CI is good or bad?

Line 196

Could a transformation of the data allow you to use a t-test also?

Line 200-202

I don’t quite understand what you are testing here or why the results are stated here? Could you please clarify.

Results

Firstly – could both figures be converted to higher resolution – a little blurry at the moment and hard to read

Line 230-231

Could you please say how much it was reduced here – given the high replication you are able to detect subtle effects but including the magnitude of the effect (as you have done for other factors such as growth) provides a clearer picture.

Discussion

The results of the study are quite clear and important. However, I think the discussion could do with some reworking in general to better put these results in the greater context of the effects of climate change in the oceans. Firstly, there tends to be a description of the findings and then a comparison with other studies for each response but not much discussion of what these findings might mean for the ecology of seaweed communities. Secondly, each response variable is mostly discussed in isolation, with little attention given to the findings as a whole. I consider that the manuscript (which has very interesting results) will substantially benefit by some reworking of the structure of the discussion and the better framing of the results in the broader context of the climate change impacts in the ocean.

My suggestions are as follows:

1. I would suggest a slight rephrasing of the first paragraph to highlight the implications of the findings – this is sort of done by the first two sentences in a general sense but it would be good to see the key results and the specific implications of them.

2. Discuss the key findings of increases in surface area, with no increase in weight but decreased resistance to breakage. This is mostly done in between lines 261-283 but it could be a little more concisely written and better placed in the context of the ecology by mentioning briefly what increased loss of seaweeds due to breakage could mean for coastal systems – e.g. habitat loss, food loss etc.

3. Discuss the findings related to carbon use to be more focused on your findings and what they mean – presently they are a little bit too focused on what others have found and don’t place your results in the context of the broader literature and are not used to explain any of your other results - e.g. an increase in growth due to CCM down-regulation.

4. Place some sub-headings for the discussion so that it is easier to follow.

Line 285

Photosystem 1? Does Fv/Fm not only measure PSII? Also it may be important to state that measuring PSII may not reflect photosynthetic rate per se as only getting half the story.

Line 292

RuBisCo is the site of carbon fixation rather than an active transport protein, can this be reworded to be more accurate.

6. PLOS authors have the option to publish the peer review history of their article (what does this mean?). If published, this will include your full peer review and any attached files.

Reviewer #1: **Yes: **Conall McNicholl

Reviewer #2: No

---

## [Author Response · Author response to Decision Letter 0]

19 Nov 2020

Dear Dr. Cornwall,

We would like to thank you for the editorial feedback on our submission PONE-D-20-25580. Based on your feedback and the reviewer’s comments, we have made changes to the manuscript and believe it to be much improved. 

We have prepared a detailed point-by-point response letter for you (with our replies in red) where we have addressed all comments from Reviewer 2 and what we believe to be non-editorial comments from Reviewer 1. Rather than refer to specific changes in the response letter for the editorial comments from reviewer 1, these have been addressed in the track-changes copy of the manuscript.

We look forward to your response to our revised version of the manuscript and hope we have satisfactorily addressed all previous issues.

Sincerely,

Alexandra Kinnby (on behalf of all authors)

Editor:

I have received the reports of two different reviewers back. They both support the work and offer minor comments that should be addressed in the revisions. I have also read the manuscript, and I consider that the authors should pay particular attention to the comments from reviewer 2 regarding the numbers of tanks etc. This is extremely important in determining the validity of the replication here. Please indicate how many header tanks, experimental tanks etc. per treatment, and how any interdependence was dealt with in the model. I cannot currently see this clearly described. 

Response: We have made numerous updates to the manuscript, all of which are included in the track changes version of the manuscript. Using the comments from both reviewer 1 and 2 we have tried to clarify our experimental array and replication structure as much as possible. In short, we used 4 header tanks for each CO2 treatment (i.e. a total of 8 header tanks) that supplied 15 aquaria each (i.e. 60 aquaria per treatment and 120 aquaria in total). We tested the effect of header tank in a mixed model ANOVA with header tank nested within CO2 treatment but found no significant effects. Therefore, the mean square of header tank was pooled with the residual mean square and we ended up using a paired t-test since the seaweed pieces in different treatments were from the same individual. We also tested the effect of header tank for controls and pCO2 treated seaweeds in two separate one-way ANOVAs with header tank as a random 4-level factor (n=15), but did not find significant differences between header tanks for any of the variables (the lowest p-value was 0.254). We chose to present the mixed-model analysis in the manuscript, which has been clarified in the statistics part of the methods section.

Reviewers' comments:

Reviewer #1: Kinnby et al exposed Fucus vesiculosus and Littorina littorea to ocean acidification conditions predicted for the year 2100 to look at the direct and indirect effects on organismal response and grazer interaction, respectively. Kelp growth, breaking strength, phlorotannin production were affected by OA, while grazer preference was unaffected but grazers decreased consumption and improved condition index. The manuscript provides new information on ecologically important species performance and interaction in future oceans. Overall, the manuscript is presented and written in a straightforward manner. 

However, more details are needed in the methods section. As the authors point out, there are discrepancies in the OA literature, much of which likely comes from different experimental setups and techniques, so it is imperative to provide as much information as possible. Provided the authors amend the methods section, I think this manuscript would be a good addition for Plos One and add to our understanding of OA effects on different species. I have performed the review with track changes in Word, pointing out missing details that should be added, also editing some text to help clarity or flow, which the authors can use at their discretion. 

Response: We thank reviewer 1 for the thorough review, which has greatly improved the clarity and flow of our manuscript! All editorial comments have been addressed using track changes. We have further detailed our experimental set-up in the methods section using comments provided in the track changes file from reviewer 1 as a guide and hope that we have clarified this. We have chosen to include our responses to specific comments below:

Line 189-196 & Table 1

How was pH calibrated? NBS? Proper standard OA TRIS buffers?

Response: pH was calibrated using NBS buffers, information about frequency of measurements as well as how alkalinity and pH was calculated has been added together with standard deviations in table 1.

Line 256-260

This is a little confusing. So, were these apical pieces of the same individual? Of the same size? Or the remaining individual? And performed at the same time or after the grazing experiment?

Response: The autogenic control containers where identical to the containers with herbivores, except for the herbivores. Since the seaweeds are alive and will grow during the experiment, it is customary to include autogenic controls in feeding preference experiment with seaweeds. We have now clarified this in the manuscript.

Line 313

How was this calculated? I am getting ~30% higher based on the means provided

Response: Thank you for pointing this out. This was calculated from the raw data, by using the cm2 value of growth (final area - initial area for each individual). The percentages are presented as a statistical tool to standardise the results by starting weight and were calculated from the same raw data. This number has now been changed in the manuscript.

Line 352-353

Was feeding quantified during the 30 d? Perhaps they ate more during the 30 d and grew more but the snails from high CO2 were stressed since the feeding experiment was performed at ambient CO2? And so during that 24 hr it seems they ate less.

Response: Feeding was only quantified during the 24 h feeding experiment and not during the 30-day exposure to CO2, so we cannot say if herbivores ate more or less during this time. It could be as the reviewer suggest, but from the design of our experiment we cannot tell.

Line 460-461

Is there anything in the literature showing low pH reduces metabolic rates?

Response: We have added a reference that shows that metabolic rate reduces at low pH in other marine invertebrates.

Line 463

Again, I would be interested in how the 30 d exposure potentially affected the shell. How would the index be affected if there was greater tissue and lower shell density? I know for some calcifying algae there has been increased organic tissue and lower CaCO3 content, not sure about for inverts

Response: The shell weight is a component of the condition index (see formula in the manuscript). We analyzed the shell weight data but found no statistically significant difference due to exposure to CO2. Therefore, we decided not to include these data in the manuscript.

Line 488-491

Reword and perhaps tone down. Avoid words such as "striking". And "substantial loss" for things that were not quantified.

Response: We agree with the reviewer and have deleted the words striking, much and substantial from the sentence.

Reviewer #2: The manuscript provides interesting and useful data on the response of habitat forming seaweeds to ocean acidification. With some minor improvements i consider the manuscript warrants publication. The main issue i have is the discussion in which i think the interesting results could be better placed in the context of the broader literature and structured to better highlight the impacts of the key findings. 

Specific comments: 

Abstract 

Line 34

change “available” to “availability” 

Response: This change has been made.

Introduction

Line 41

in surface and coastal waters? A little confusing with the wording, I think you mean to suggest that its surface of the open ocean but the coastal waters will be all the water column due to it being shallow. Might help to have a reword for clarity. 

Response: This sentence has been reworded for clarity.

Line 42-43

The sentence that says coastal organisms the most impacted. I suggest saying one of the most impacted. 

Response: Addressed

Line 55-56

See also van der Loos et al 2019 for no change. 

Response: This reference has been added.

Line 83

Change “Temporal” to “Temperate” 

Response: This change has been made.

Line 90-94

This sentence is a little confusing – suggest a slight reword. 

Response: This sentence has been reworded for clarity.

Line 94-95

I have noticed a little bit of repeatability about the importance of fucoids/seaweeds in general in coastal systems – e.g. Lines 83- 84 and 43-44. I suggest going through the introduction and reducing repetition of this point wherever possible. 

Response: This has been addressed, we have tried to minimize repetition as much as possible.

Line 96-99

Suggest framing this point as important rather than interesting as it is both interesting and important! 

Response: This has been reworded.

Line 101-107

Are both these species common globally or just in the region you work? As written, it suggests globally. 

Response: This has been clarified in the manuscript.

Methods 

Line 128-129

Could you provide frequency and total number of measurements? Also what equipment was used to measure these parameters? This info needs to be included. 

Response: This information has been added to the manuscript. The pCO2 was checked daily, however the values in table 1 are based on biweekly measurements (n=8).

Table 1

Please provide standard error in the table for the measured parameters. Was AT estimated from only pCO2 and salinity or also pH and temp? Needs to be clarified in the caption/text. Was pH measured on the NBS scale only? Ideally seawater pH should be measured on the Total scale. For comparison to other studies could the pH also be reported on the Total scale (if not measured) using estimates from CO2Calc and this stated? 

Response: This information has been added in the caption, the table, and in the text. Total alkalinity was estimated from salinity using a long-term salinity:alkalinity relationship for this location (r = 0.94). This method has been shown to generate very low uncertainties in estimates of carbonate system parameters (uncertainties ≤±0.006 pHNBS and ±0.08 ΩAr) (for more details see reference 38 in the manuscript, Eriander et al., 2016). pHT was calculated in CO2Calc using temperature, salinity, total alkalinity, and pCO2 data.

Line 135

All seaweed thalli? Same question for phlorotannin and elemental analysis and breaking strength. Please clarify and include n per treatment for each response variable.

Response: This information, including sample sizes, has been clarified.

Line 140

Could you include make and model of the equipment.

Response: This information has been added to the manuscript. 

Line 148

Change to “as a standard”. 

Response: This has been changed.

Line 157-177

I found it a little hard to follow whether the snails exposed to different pCO2 were placed in separate containers and then also if not how this was accounted for in the data analysis? Some more clarity would be useful.

Response: Information has been added to this section for clarity.

Line 179

For readers not familiar with condition index could you provide a brief description of whether higher or lower CI is good or bad?

Response: Information has been added for clarity.

Line 196

Could a transformation of the data allow you to use a t-test also?

Response: Transforming the data does not make it normally distributed. However, this data has now been analyzed with a Mann-Whitney U-test as this seems more fitting. Nevertheless, this does not change the results and conclusions.

Line 200-202

I don’t quite understand what you are testing here or why the results are stated here? Could you please clarify. 

Response: This text is not a statistical test or describing results; it is included to make it easier for the reader to interpret the results that are presented in the results-section. Depending on how the difference between wet weight changes is constructed, the results will mean different things. The way we constructed the difference, a result that show a significantly lower difference in wet weight change for seaweed pieces kept with herbivores will mean a preference for feeding on control seaweeds. If we had constructed it the other way around, it would have meant a preference for the treated seaweeds. We have now clarified this point. For more information on how to analyze feeding preference experiments, please refer to the publication by Peterson and Renaud (1989).

Results

Firstly – could both figures be converted to higher resolution – a little blurry at the moment and hard to read 

Response: The figures have been converted, hopefully to satisfactory resolution.

Line 230-231

Could you please say how much it was reduced here – given the high replication you are able to detect subtle effects but including the magnitude of the effect (as you have done for other factors such as growth) provides a clearer picture.

Response: This has been added to the manuscript.

Discussion

The results of the study are quite clear and important. However, I think the discussion could do with some reworking in general to better put these results in the greater context of the effects of climate change in the oceans. Firstly, there tends to be a description of the findings and then a comparison with other studies for each response but not much discussion of what these findings might mean for the ecology of seaweed communities. Secondly, each response variable is mostly discussed in isolation, with little attention given to the findings as a whole. I consider that the manuscript (which has very interesting results) will substantially benefit by some reworking of the structure of the discussion and the better framing of the results in the broader context of the climate change impacts in the ocean. 

My suggestions are as follows: 

My suggestions are as follows:

1. I would suggest a slight rephrasing of the first paragraph to highlight the implications of the findings – this is sort of done by the first two sentences in a general sense but it would be good to see the key results and the specific implications of them. 

Response: This paragraph has been altered with this suggestion in mind. 

2. Discuss the key findings of increases in surface area, with no increase in weight but decreased resistance to breakage. This is mostly done in between lines 261-283 but it could be a little more concisely written and better placed in the context of the ecology by mentioning briefly what increased loss of seaweeds due to breakage could mean for coastal systems – e.g. habitat loss, food loss etc. 

Response: More information has been added to discuss the effects of these results in a broader context.

3. Discuss the findings related to carbon use to be more focused on your findings and what they mean – presently they are a little bit too focused on what others have found and don’t place your results in the context of the broader literature and are not used to explain any of your other results - e.g. an increase in growth due to CCM down-regulation. 

Response: We have made changes to this paragraph with this comment in mind.

4. Place some sub-headings for the discussion so that it is easier to follow. 

Response: Sub-headings have been added to the discussion.

Line 285

Photosystem 1? Does Fv/Fm not only measure PSII? Also it may be important to state that measuring PSII may not reflect photosynthetic rate per se as only getting half the story. 

Response: Thank you for pointing this out. The manuscript has been modified accordingly.

Line 292

RuBisCo is the site of carbon fixation rather than an active transport protein, can this be reworded to be more accurate. 

Response: RuBisCo has been taken out from this sentence.

---

## [Decision Letter · Decision Letter 1]

16 Dec 2020

PONE-D-20-25580R1

Ocean acidification decreases grazing pressure but alters morphological structure in a dominant coastal seaweed

PLOS ONE

Dear Dr. Kinnby,

Thank you for submitting your manuscript to PLOS ONE. After careful consideration, we feel that it has merit but does not fully meet PLOS ONE’s publication criteria as it currently stands. Therefore, we invite you to submit a revised version of the manuscript that addresses the points raised during the review process.

I have received both reviews back. Only one reviewer indicates that there are minor revisions required, while they other indicated accept as is. Please make these minor adjustments and resubmit your manuscript. After these are made I will accept the manuscript.

We look forward to receiving your revised manuscript.

Kind regards,

Christopher Edward Cornwall, Ph.D.

Academic Editor

PLOS ONE

Reviewers' comments:

Reviewer's Responses to Questions

**Comments to the Author**

1. If the authors have adequately addressed your comments raised in a previous round of review and you feel that this manuscript is now acceptable for publication, you may indicate that here to bypass the “Comments to the Author” section, enter your conflict of interest statement in the “Confidential to Editor” section, and submit your "Accept" recommendation.

Reviewer #1: (No Response)

Reviewer #2: All comments have been addressed

2. Is the manuscript technically sound, and do the data support the conclusions?

Reviewer #1: Yes

Reviewer #2: Yes

3. Has the statistical analysis been performed appropriately and rigorously? 

Reviewer #1: Yes

Reviewer #2: Yes

4. Have the authors made all data underlying the findings in their manuscript fully available?

Reviewer #1: Yes

Reviewer #2: Yes

5. Is the manuscript presented in an intelligible fashion and written in standard English?

Reviewer #1: Yes

Reviewer #2: Yes

6. Review Comments to the Author

Reviewer #1: Kinnby et al. have added more details to the method section and cleared up other uncertainties as requested. I just have some minor suggestions for further improvement of the article. I think with these edits the manuscript will be ready for publication.

14: insert “projected to be 0.4 units lower..”

46: recommend changing “some” to “many” or even most

66: insert pCO2 at the end of the sentence to give reference to the concentrations you are talking about

72: here and a couple other places, OA is spelled out

87: remove “very”

88: since they may not be carbon limited

117: You should report the light levels if measured and type of light? LED? Natural? This is important since you discuss photosynthetic implications

128: space between number and units

141: change sd to SD

176: add parentheses after SD & change all sd to SD

251-252: Combine these two sentences… “The stable carbon isotope content of F. vesiculosus was significantly reduced to -13% when exposed to elevated pCO2.”

276 -278: reword, “which” is used twice in this sentence

276-281: These last sentences could use some work, its difficult to read. Maybe switch the last one around, “While the condition index increased…there was no change in preference and consumption actually decreased”

291: But since you used genetically identical samples, how does this new information add to our existing understanding?

295: See Guenther et al 2017 “Macroalgal spore dysfunction” I believe they show weakened attachment strength – which could add to your story

325: CCM - write out in full first time

327: Species names should be written out in full at the beginning of a sentence

371: reword – suggest "Reduced consumption for the herbivore…”

373: Was it really unanticipated?

Reviewer #2: (No Response)

7. PLOS authors have the option to publish the peer review history of their article (what does this mean?). If published, this will include your full peer review and any attached files.

Reviewer #1: No

Reviewer #2: No

---

## [Author Response · Author response to Decision Letter 1]

18 Dec 2020

Response to reviewers:

Reviewer #1: Kinnby et al. have added more details to the method section and cleared up other uncertainties as requested. I just have some minor suggestions for further improvement of the article. I think with these edits the manuscript will be ready for publication.

14: insert “projected to be 0.4 units lower..”

Response: The word lower has been inserted.

46: recommend changing “some” to “many” or even most

Response: Some has been changed to most.

66: insert pCO2 at the end of the sentence to give reference to the concentrations you are talking about

Response: pCO2 has been added to the end of the sentence.

72: here and a couple other places, OA is spelled out

Response: Thank you for pointing this out, this has been changed here and throughout the manuscript.

87: remove “very”

Response: Very has been removed from the sentence.

88: since they may not be carbon limited

Response: “Since they may not be carbon limited” has been added to the end of the sentence.

117: you should report the light levels if measured and type of light? LED? Natural? This is important since you discuss photosynthetic implications

Response: The experiment was performed in a greenhouse with natural lighting (natural light cycle 18:6 h, L:D).

128: space between number and units

Response: A space has been added.

141: change sd to SD

Response: sd has been changed to SD

176: add parentheses after SD & change all sd to SD

Response: A parentheses has been added and sd has been changed to SD throughout the manuscript.

251-252: Combine these two sentences…”The stable carbon isotope content of F. vesiculosus was significantly reduced to -13% when exposed to elevated pCO2.

Response: These two sentences have been merged to one.

276-278: reword, “which” is used twice in this sentence

Response: This sentence has been modified.

276-281: These last sentences could use some work, it is difficult to read. Maybe switch the last one around, “While the condition index increased…there was no change in preference and consumption actually decreased”

Response: These last sentences have been reworked.

291: But since you used genetically identical samples, how does this new information add to our existing understanding?

Response: This sentence has been altered for clarity. We want to clarify to the reviewer that we used the same individual in the treatment and its control, but there were not genetically identical samples for the entire experiment.

295: See Guenther et al 2017 “Macroalgal spore dysfunction” I believe they show weakened attachment strength – which could add to your story

Response: Thank you for bringing this article to our attention, we have added it to the text.

325: CCM – write out in full first time

Response: “carbon dioxide-concentrating mechanism” has been added to the sentence

327: Species names should be written out in full at the beginning of a sentence

Response: The full species name has been written out.

371: reword – suggest “Reduced consumption for the herbivore…”

Response: This has been reworded.

373: Was it really unanticipated?

Response: Yes, we weren’t expecting to find that the seaweeds would break more easily after exposure to elevated levels of pCO2. To our knowledge the literature on fleshy algae is pointing towards a positive effect of OA on these species since it generally enhances growth. As far as we know this is the first study that suggests that despite OA having a positive effect on growth there is a synergistic effect resulting in the algae becoming more vulnerable. Based on the available literature we did not expect to find this effect, and so it was unexpected for us. We appreciate that the reviewer brought our attention to Guenther et al. (2017) who suggest a similar pattern for spore attachment in red seaweeds.

Reviewer #2: (No Response)

---

## [Editor Report · Decision Letter 2]

21 Dec 2020

Ocean acidification decreases grazing pressure but alters morphological structure in a dominant coastal seaweed

PONE-D-20-25580R2

Dear Dr. Kinnby,

We’re pleased to inform you that your manuscript has been judged scientifically suitable for publication and will be formally accepted for publication once it meets all outstanding technical requirements.

Kind regards,

Christopher Edward Cornwall, Ph.D.

Academic Editor

PLOS ONE
---

## [Editor Report · Acceptance letter]

6 Jan 2021

PONE-D-20-25580R2 

Ocean acidification decreases grazing pressure but alters morphological structure in a dominant coastal seaweed 

Dear Dr. Kinnby:

I'm pleased to inform you that your manuscript has been deemed suitable for publication in PLOS ONE. Congratulations! Your manuscript is now with our production department. 

Kind regards, 

on behalf of

Dr. Christopher Edward Cornwall 

Academic Editor

PLOS ONE